# Increased *SEC23A* Expression Correlates with Poor Prognosis and Immune Infiltration in Stomach Adenocarcinoma [note 1]

**DOI:** 10.3390/cancers15072065

**Published:** 2023-03-30

**Authors:** Su Zhaoran, Christina Susanne Linnebacher, Michael Linnebacher

**Affiliations:** 1Departments of Gastrointestinal Surgery, People’s Hospital of Tongling City, Tongling 244000, China; 2College of Mathematics and Computer Science, Tongling University, Tongling 244000, China; 3Molecular Oncology and Immunotherapy, Clinic of General Surgery, University Medical Center Rostock, 18057 Rostock, Germany; 4Patient Models for Precision Medicine, Clinic of General Surgery, University Medical Center Rostock, 18057 Rostock, Germany

**Keywords:** stomach adenocarcinoma, *SEC23A*, immune infiltration, biomarker, prognosis

## Abstract

**Simple Summary:**

Research on the biological and molecular characteristics of stomach adenocarcinoma (STAD) is mandatory to identify molecular markers and targets for diagnosis, prognosis and therapeutic interventions. The *SEC23A* gene is involved in the occurrence and development of various tumor entities. However, little is known about its expression and relevance for STAD. By combining computational biology with validation on patient tissue samples, this study is the first to describe the significantly upregulated expression of *SEC23A* in STAD. We also identified an association with disease progression, STAD patients’ prognosis, and several infiltrating immune cell types and their activity. We observed the significantly upregulated expression of *SEC23A* in STAD, an association with disease progression, patients’ prognosis and infiltrating immune cell subsets. Thus, we propose *SEC23A* as an independent prognostic factor with a putative role in immune response regulation in STAD.

**Abstract:**

Background: Previous studies have described that the *SEC23A* gene is involved in the occurrence and development of various tumor entities. However, little is known about its expression and relevance in stomach adenocarcinoma (STAD). The aim of this study was to bioinformatically analyze the role of *SEC23A* in STAD, followed by patient tissue sample analyses. Materials and methods: *SEC23A* expression levels in STAD and normal gastric tissues were analyzed in the Cancer Genome Atlas and Gene Expression Omnibus databases; results were verified in fresh clinical STAD specimens on both gene and protein expression levels. *SEC23A* expression correlated with survival parameters by Kaplan–Meier and multivariate Cox regression analyses. The top genes co-expressed with *SEC23A* were identified by gene set enrichment analysis (GSEA) using the clusterProfiler package in R. Furthermore, the R package (immunedeconv), integrating the CIBERSORT algorithm, was used to estimate immune cell infiltration levels in STAD. Results: *SEC23A* gene and sec23a protein expression were both significantly upregulated in STAD, and this correlated with the pT stage. Moreover, high *SEC23A* expression was associated with poor disease-free and overall survival of STAD patients. Cox analyses revealed that besides age and pathologic stage, *SEC23A* expression is an independent risk factor for STAD. GSEA indicated that *SEC23A* was positively associated with ECM-related pathways. In the CIBERSORT analysis, the level of *SEC23A* negatively correlated with various infiltrating immune cell subsets, including follicular helper T cells, Tregs, activated NK cells and myeloid dendritic cells. Finally, the expression levels of immune checkpoint-related genes, including HAVCR2 and PDCD1LG2, were significantly increased in the high *SEC23A* expression group. Conclusions: We observed the significantly upregulated expression of *SEC23A* in STAD, an association with disease progression, patients’ prognosis and infiltrating immune cell subsets. Thus, we propose *SEC23A* as an independent prognostic factor with a putative role in immune response regulation in STAD.

## 1. Introduction

Stomach adenocarcinoma (STAD) is one of the most common digestive tract malignant tumors in clinics [1]. China has one of the highest STAD incidence rates. More alarmingly, morbidity and mortality are twice as high as the world average. Surveys have shown that the incidence and mortality of STAD in China ranks second and third for malignant tumors [2]. The 5-year survival rate of early STAD after radical treatment can surpass 90% [3]. However, due to unspecific symptoms in the early stages, the lack of routine gastroscopic physical examination, and frequently patient awareness, more than 80% of STAD patients have progressed at the time of diagnosis, resulting in a very poor prognosis. Therefore, further research on the biological and molecular characteristics of STAD is mandatory to identify molecular markers and targets for diagnosis, prognosis and therapeutic interventions.

The secretome of tumor cells plays an important role in the mechanism of metastatic niche formation. It influences the adaption of circulating tumor cells in the unfamiliar microenvironment when reaching distant locations [4]. The first step in biosynthetic secretion, the exit of proteins from the endoplasmic reticulum, is mediated by coat protein complex II (COPII). COPII forms the coat protein of endoplasmic reticulum secretory vesicles, which are responsible for transport to the Golgi apparatus [5]. The two isoforms of the SEC homologous protein (sec23), sec23a and sec23b, both contain five distinct functional domains (i.e., α-helix, β-barrel, gelsolin domain, trunk domain, and zinc finger). They are members of the sec23/sec24 protein family with an important role in the assembly of COPII [6], which is composed of several large protein subunits, mainly including sec23/sec24 and sec13/sec31 dimers.

Previous research has demonstrated that *SEC23B* mutations may cause congenital dyserythropoietic anemia type II (CDAII) [7], while the mutations of *SEC23A* may cause cranio-lenticulo-sutural dysplasia (CLSD) [8]. In vivo, the functions of *SEC23A* and *SEC23B* have been found to be interchangeable in COPII, whereas *SEC23A* and *SEC23B* may have the opposite activity in human cancer for unknown reasons [6]. Moreover, recent studies have confirmed that the sec23a protein triggers the occurrence and development of various tumors [9,10,11,12]. However, few studies have analyzed its expression level and potential involvement in STAD development.

Therefore, we investigated the expression and epigenetic regulation of *SEC23A* in STAD, its possible pathogenic mechanism, and its correlation with prognostic parameters by combining bioinformatics with expression analyses in fresh tissue samples.

## 2. Materials and Methods

### 2.1. Expression Analyses of SEC23A by Bioinformatics

Xiantao is a comprehensive bioinformatics tool (https://www.xiantao.love (accessed on 20 June 2022) based on R language that has realized online analysis and visualization through front–end technology to conduct analysis of public database data [13]. We first applied this tool to The Cancer Genome Atlas (TCGA, https://portal.gdc.cancer.gov/ (accessed on 20 June 2022)) and the Genotype-Tissue Expression (GTEx, gtexportal.org/home/ (accessed on 20 June 2022)) databases to analyze the mRNA levels of *SEC23A* in 33 cancer entities. Then, two STAD datasets, GSE54129 [14] and GSE118916 [15], were downloaded from the GEO database and analyzed to verify the expression pattern of *SEC23A* in STAD. Details of the flow chart are provided in Appendix A.

### 2.2. Tissue Samples and Real-Time Fluorescence Quantitative PCR (qRT–PCR) Analysis

We collected pairs of treatment-naïve human STAD and adjacent normal gastric tissues from 16 patients who underwent surgery at the People’s Hospital of Tongling City. There were 10 males and 6 females among the 16 patients, with an average age of 68.9 years. According to the 8th edition of UICC staging, there was: 1 patient with stage I, 9 with stage II and 6 with stage III. Seven cases underwent laparoscopic-assisted total gastrectomy and 9 cases underwent laparoscopic-assisted distal gastrectomy. This investigation was approved by the Ethics Committee of the People’s Hospital of Tongling City (approval no. 2022003), and all patients signed informed consent forms. Samples were stored at −80 °C prior to mRNA isolation with TRIzol (Thermo Fisher Scientific, Waltham, MA, USA) and reverse transcription to cDNA (iScript Bio-Rad, Shanghai, China) for genetic expression analyses. Real-time qRT–PCR was performed on the PIKOREAL 96 (Thermo Fisher Scientific) using FastStart Universal SYBR Green Master mix (Merck, Darmstadt, Germany) and β-actin as internal reference. The gene-specific primers used were: *SEC23A* (forward: 5′-TGGTTGGAGATGAGTTGAAG-3′; reverse: 5′-AGTTGTAGCAGCTCGATTAG-3′) and *β-actin* (forward: 5′-CCCTGGAGAAGAGCTACGAG-3′; reverse: 5′-GGAAGGAAGGCTGGAAGAGT-3′). Data analysis was performed using the 2^−ΔΔCt^ method.

### 2.3. Immunohistochemistry and Western Blot Assay

Western blot assays were carried out as previously described with antibodies against sec23a (AD3215602-7, Abcam, Boston, MA, USA, 1:10,000 dilution) and β-actin (19C10509, Zsbio, Beijing, China, 1:1000 dilution). Immunohistochemical staining of sec23a (AD3215602-7, Abcam) was performed at a concentration of 1:100 and scoring was performed independently by two pathologists. Automated scores of the Western blots and immunohistochemical staining slides were obtained using Image J software (National Institutes of Health, Bethesda, MD, USA) and used to compare tumor versus normal tissues [16].

### 2.4. Analysis of SEC23A Expression with Clinicopathological Features and Survival

Details of the flow chart are provided in Appendix A. Datasets of 407 STAD patients, including clinical information and gene expression values, were downloaded from the TCGA database. STAD patients were grouped according to the variables: sex, age, pathological stage, tumor stage (pT), lymph node status (pN) and metastasis (pM) for univariate analysis of the expression of *SEC23A*. Valid prognostic information and *SEC23A* expression data were available for 370 STAD patients. The log-rank method was used for survival analysis to correlate *SEC23A* expression (based on the median) with overall survival (OS), disease-specific survival (DSS) and progression-free interval (PFI). Cox regression analysis was used to determine the risk factors for OS in 347 patients with complete background data.

### 2.5. Analyses of Genes Co-Expressed with SEC23A and Gene Set Enrichment Analysis (GSEA)

The GEPIA2 online database tool (http://gepia2.cancer-pku.cn/ (accessed on 20 June 2022)) was used to estimate the top 100 genes co-expressed with *SEC23A* in STAD. Enrichment analysis was performed using the clusterProfiler package in R (version 4.0.3).

GSEA for TCGA-STAD data was done using the GSEA v4.3.0 software. The file c2.cp.v7.2.symbols.gmt was selected for further analysis. The number of permutations was 5000, and the cut-offs for significant enrichment were a normalized enrichment score >3, a false discovery rate q-val <0.05 and an adjusted *p* value (p.adjust) < 0.05.

### 2.6. Evaluation of Tumor-Infiltrating Immune Cells and Expression of Checkpoint-Related Genes

For estimation of immune infiltration in STAD, the R package (immunedeconv) was utilized to integrate CIBERSORT, which is a deconvolution algorithm based on gene expression that can evaluate the changes in the expression levels of one set of genes relative to all other genes in the sample. The abundances of 22 types of immune cells (B-cell naive, B-cell memory, B-cell plasma, T-cell CD8+, T-cell CD4+ naive, T-cell CD4+ memory resting, T-cell CD4+ memory activated, T-cell follicular helper, T-cell regulatory (Tregs), T-cell gamma delta, NK-cell resting, NK-cell activated, Monocyte, Macrophage M0, Macrophage M1, Macrophage M2, Myeloid dendritic cell resting, Myeloid dendritic cell activated, Mast cell activated, Mast cell resting, Eosinophil and Neutrophil) were estimated. Briefly, gene expression datasets were uploaded to the Xiantao bioinformatics tool after standard annotation. The immunedeconv R package estimated a P value for deconvolution via the CIBERSORT algorithm. Then, the differential expression of immune checkpoint-related genes, including *CD274*, *CTLA4*, *HAVCR2*, *LAG3*, *PDCD1*, *PDCD1LG2*, *TIGIT* and *SIGLEC15*, between the high and low *SEC23A* expression groups in STAD based on the TCGA gene expression data was estimated.

### 2.7. Statistical Methods

SPSS 19.0 and R 4.0.3 software programs were used to perform the analysis. The independent sample *t* test, paired sample *t* test and Mann–Whitney U test were used for the comparison of the two groups. Log-rank and multivariate Cox regression analyses were used to study the prognosis, and Spearman analysis was used to assess the relationship between *SEC23A* expression and co-expressed genes. A *p* value < 0.05 was considered to be statistically significant.

## 3. Results

### 3.1. Transcriptional Levels of SEC23A in Pan-Cancer and STAD

By bioinformatically analyzing the expression levels of *SEC23A* among various cancers in TCGA and GTE-x, the transcriptional levels of *SEC23A* were found to be divergent in 33 tumors (Figure 1A). As shown in Figure 1B, the expression levels of *SEC23A* were significantly increased in STAD compared to normal tissues. Similarly, *SEC23A* mRNA expression was upregulated in two STAD datasets of the GEO database (Figure 1C,D). These results could subsequently be reproduced by qRT–PCR analysis of 16 pairs of freshly collected STAD and adjacent normal human tissue samples. Here, *SEC23A* expression was also significantly increased in the cancer tissues (Figure 1E).

### 3.2. sec23a Protein Was Upregulated in Clinical STAD Specimens

Subsequently, sec23a protein expression was investigated by Western blot assays and immunohistochemistry analyses using freshly collected 16 paired STAD and adjacent normal tissues. Indeed, the sec23a protein was expressed at higher levels in cancer tissues compared to the adjacent normal tissues both in Western blot (Figure 1F: gray values were for tumor 0.64 ± 0.08 and for normal 0.12 ± 0.02, *p* < 0.001) and immunohistochemistry (Figure 1G: the percentage of positive and high positive stained area combined was for tumor 66.4 ± 18.2 and for normal 37.8 ± 12.6, *p* < 0.001).

### 3.3. Association between SEC23A Expression and Clinicopathological Variables

Next, the normalized expression of *SEC23A* of the 407 available TCGA database STAD patients was compared according to groups for: sex (male vs. female; 4.67 ± 0.67 vs. 0.12 ± 0.02), age (<=65 vs. >65; 4.68 ± 0.70 vs. 4.69 ± 0.68), UICC staging (I vs. II vs. III vs. IV; 4.55 ± 0.80 vs. 4.61 ± 0.63 vs. 4.75 ± 0.65 vs. 4.80 ± 0.69), as well as separately to pT (1 vs. 2 vs. 3 vs. 4; 4.14 ± 0.53 vs. 4.64 ± 0.75 vs. 4.60 ± 0.63 vs. 4.91 ± 0.65), pN (0 vs. 1 vs. 2 vs. 3; 4.62 ± 0.71 vs. 4.60 ± 0.64 vs. 4.75 ± 0.68 vs. 4.76 ± 0.68) and pM (0 vs. 1; 4.68 ± 0.67 vs. 4.70 ± 0.85). There, only pT was significantly correlated in univariate analysis (Figure 2).

### 3.4. High SEC23A Expression Predicted Poor Prognosis in GC Patients

The 370 STAD patients with valid prognostic information and *SEC23A* expression data obtained from the TGCA dataset subsequently underwent survival analysis based on the *SEC23A* median expression value. According to Kaplan–Meier analysis (Figure 3A–C), increased *SEC23A* expression was significantly correlated with poor OS (*p* = 0.010) and DSS (*p* = 0.027). We further performed Cox regression analysis on 347 STAD patients, including OS, *SEC23A* expression and clinical parameters. In univariate analysis, age > 65 years, higher tumor stages, distant metastasis, positive lymph node status (N1 and N3), and the expression of *SEC23A* (median based) were significantly associated with worse OS (Table 1 and Figure 3D). In the multivariate analysis, however, only high *SEC23A* expression (*p* = 0.041) and age > 65 years (*p* = 0.001) remained independent prognostic factors for poor OS.

### 3.5. Analyses of Genes Co-Expressed with SEC23A in STAD

The top 100 genes co-expressed with *SEC23A* were selected to conduct an enrichment analysis. The terms “muscle contraction”, “muscle system process” and “cell-matrix adhesion” were significantly enriched in the GO biological process analysis (Figure 4A and Table 2). The terms “contractile fiber part”, “myofibril” and “contractile fiber” were significantly enriched in the GO term cellular component analysis (Figure 4A and Table 2). According to the molecular function analysis, the terms “actin binding”, “dystroglycan binding” and “tubulin binding” were highly enriched (Figure 4A and Table 2). KEGG pathway analysis indicated that the “cGMP-PKG signaling pathway”, “arrhythmogenic right ventricular cardiomyopathy” and “hypertrophic cardiomyopathy” were significantly enriched (Figure 4A and Table 2). The top six co-expressed genes of *SEC23A* in STAD arranged by adjusted *p* values were *KCTD10*, *CORO1C*, *ZYG11B*, *RBFOX2*, *PARVA* and *LAMA4* (Figure 4B).

### 3.6. GSEA Identified SEC23A-Related Pathways

GSEA was conducted between the high and low *SEC23A* expression groups from the TCGA-STAD dataset to analyze the possible biological pathways regulated by *SEC23A* in STAD. A total of 81 pathways were significantly enriched in the *SEC23A* high expression group, with the top 9 set size terms being: “pid_integrin1_pathway”, “kegg_ecm_receptor_interaction”, “naba_ecm_glycoproteins”, “reactome_non_integrin_membrane_ecm_interactions”, “reactome_ecm_proteoglycans”, “reactome_integrin_cell_surface_interactions, reactome_laminin_interactions”, “naba_basement_membranes” and “kegg_dilated_cardiomyopathy” (Figure 5).

### 3.7. Expression of Tumor-Infiltrating Immune Cells and Checkpoint-Related Genes

A total of 407 STAD samples were classified into high- and low-expression groups according to the median expression value of *SEC23A*. We analyzed the abundance of 22 infiltrating immune cell subsets between these two groups using the CIBERSORT algorithm. The levels of follicular helper T cells, Tregs, activated NK cells and resting myeloid dendritic cells were significantly higher, while the level of M2 macrophages was significantly lower in the low compared to the high *SEC23A* expression group (Figure 6A). In contrast, the expression levels of immune checkpoint-related genes, including *HAVCR2* (*p* < 0.05) and *PDCD1LG2* (*p* < 0.001), were significantly higher in the high compared to the low *SEC23A* expression group (Figure 6B).

## 4. Discussion

Sec23a is an important member of the sec23 protein family, and it forms heterodimers with sec24c, sec16a and sec16b, which assemble into the outer coat protein COPII. This protein complex can encapsulate protein-secreting vesicles in the endoplasmic reticulum, which are subsequently transported to the Golgi apparatus [5,17,18]. Inhibition of *SEC23A* expression reduces its protein translation level, thus primarily affecting the assembly of COPII, but also, more generally, the protein secretion of tumor cells, and can thereby contribute to a reshaping of the tumor microenvironment. In line with this, the *SEC23A* gene was shown to be involved in the regulation of metastases of different tumor types [19,20,21,22]; however, its expression and activity have not yet been elucidated in STAD.

To the best of our knowledge, this is the first study to investigate the role of *SEC23A* in STAD. Our findings in human tissue samples and public databases revealed a significantly higher expression of *SEC23A* in STAD than in tumor adjacent normal tissues. *SEC23A* expression was positively correlated with the tumor’s pT stage. The significance level was lost for the TNM staging in total, but a trend of higher staging with increasing *SEC23A* expression was still seen. Furthermore, increased *SEC23A* expression was significantly correlated with poor OS and DSS of STAD patients. The Cox regression analysis results even implied an independent prognostic factor of *SEC23A* for STAD. Combined, these findings suggest that *SEC23A* can serve as an interesting novel prognostic biomarker in STAD, similar to what has previously been suggested for bladder cancer. Zeng et al. found *SEC23A* to be an independent prognosticator for bladder cancer through biological information analysis and functional in vitro verification [9]. This was the first study to identify the oncogenic potential of *SEC23A* likely mediated through MAPK signaling. Previously, only a tumor suppressive function of *SEC23A* had been reported in prostate, breast and colorectal cancer by activating insulin-like growth factor binding protein 4 [10,11,12]. Although we did not confirm whether in STAD the pro-tumoral activity of *SEC23A* is also executed via MAPK signaling, our results strengthen the double-edged sword character of *SEC23A* for human cancer.

Co-expressed genes often have similar functions. *SEC23A* expression correlated strongly positively with *KCTD10*, *CORO1C*, *ZYG11B*, *RBFOX2*, *PARVA* and *LAMA4*. This list of genes includes confirmed candidates capable of triggering cancer development [23,24,25,26,27,28,29]. This in turn suggests that *SEC23A* itself might also act as a tumor-promoting gene, at least in STAD. To explore the underlying biological mechanisms, we performed GO, KEGG and GSEA analyses of genes co-expressed with *SEC23A*. In GO biological process analysis, we noticed “cell-matrix adhesion” as significantly enriched, which is in line with a key role for tumor progression, metastization and drug resistance [30]. According to the GO molecular function analysis, “actin binding” and “tubulin binding” were also significantly enriched. Jia et al. recently discovered the involvement of actin-binding protein in STAD progression and suggested it as a promising therapeutic target based on cell and animal experiments [31].

Using KEGG analyses, the cGMP-PKG signaling pathway was found to be significantly enriched. Taking into account that Tian et al. reported that *H*. *pylori* infection depends on ZEB1 to upregulate PRTG, which in turn activates the cGMP/PKG signaling pathway, ultimately triggering STAD development [32], this might be a very interesting direction for future research.

GSEA enrichment analysis showed that *SEC23A* was positively related to ECM-related pathways. The ECM is an acellular component and in the case of the tumor microenvironment, it provides biochemical components and basic structural support for tumor cells [33,34]. It is composed of collagen, proteoglycans, laminin and network connection proteins. The ECM is not only an intercellular filler but also an active substance facilitating intercellular communication, ultimately driving cell proliferation and adhesion. Relevant studies have confirmed that tumor ECM fibrogenesis forms a cross-linked network structure, supporting and nourishing tumor cells to allow for further tumor growth and infiltration of surrounding tissues [35,36,37]. Therefore, we consider it likely that the high expression of *SEC23A* contributes to the poor prognosis of STAD patients via the above outlined mode of action.

Another important finding in our study was the correlation between *SEC23A* expression and the level of immune cell infiltration in STAD. The CIBERSORT analysis revealed that in low *SEC23A* expressing STAD cases, the levels of Th cells, Tregs, activated NK cells and resting myeloid dendritic cells were significantly higher, while M2 macrophages were significantly fewer compared to high expressing cases. The host immune response is involved in the whole process of tumor development and growth [38], and tumor-infiltrating lymphocytes (TILs) are often interpreted as host protective factors against tumor development [39]. Matured and activated lymphocytes recruited to a tumor have enormous potential to inhibit tumor growth. Previous studies demonstrated that in patients with radically resected colon cancer, TILs at the infiltrating edge and CD8^+^ T lymphocyte density at the central site were positive prognostic factors. The DFS of patients with high CD8^+^ T lymphocyte density was significantly higher than that of patients with low density [40,41,42]. We further found significantly higher expression levels of the immune checkpoint-related genes *HAVCR2* and *PDCD1LG2* in the high *SEC23A* expression group compared to the low *SEC23A* expression group in STAD. Since expression of immune checkpoint molecules in tumor cells is regularly found to be associated with TIL levels [43], it is tempting to speculate that there might be a functional connection between expression of COPII genes like *SEC23A,* immune checkpoint-related genes and the levels of tumor-infiltrating immune cell subsets.

One of the limitations of the present study is that we did not unravel the molecular mechanisms of such connections. However, we delivered sufficient evidence to justify future studies addressing these mechanisms. Similarly, we verified the initial bioinformatics results of *SEC23A* expression only in a small cohort of tumor samples. Although we expanded this from gene expression toward protein expression, conclusions on the precise role of *SEC23A* in tumor biology and especially direct or indirect effects on immune cells in STAD need further analyses, with a focus on cellular assays.

## 5. Conclusions

By combining computational biology with validation on patient tissue samples, this study is the first to describe the significantly upregulated expression of *SEC23A* in STAD. We also identified an association with disease progression, STAD patients’ prognosis, and several infiltrating immune cell types and their activity. Therefore, we suggest that *SEC23A* represents not only an independent prognostic factor but may also have a role in regulating immune cell infiltration in STAD and might even be a predictive biomarker for immunotherapeutic interventions.

## Figures and Tables

**Figure 1 cancers-15-02065-f001:**
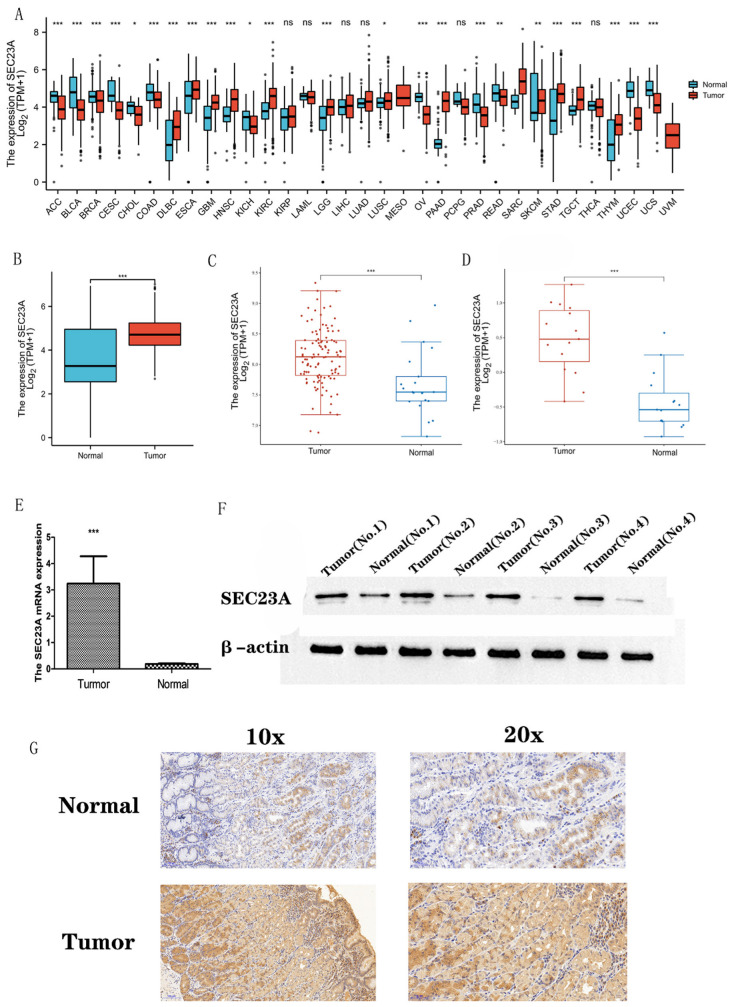
Expression of *SEC23A* in tumors (* *p* < 0.05, ** *p* < 0.01, *** *p* < 0.001, ns (not significant) *p* > 0.05). (**A**) Transcript levels of *SEC23A* in 33 cancer entities based on TCGA and GTEx. (**B**) Transcript levels of *SEC23A* in STAD based on TCGA and GTEx. (**C**) Analysis of *SEC23A* expression in tumor and normal tissues based on GSE66229. (**D**) Analysis of *SEC23A* expression in tumor and normal tissues based on GSE118916. (**E**) qRT-PCR analysis of *SEC23A* expression in matching tumor and normal tissues from 16 STAD patients. (**F**) Expression of sec23a protein analyzed by WB in matching tumor and normal tissues from 16 STAD patients. (**G**) Expression of sec23a protein analyzed by immunohistochemistry in matching tumor and normal tissues from 16 STAD patients. The uncropped blots are shown in Appendix A.

**Figure 2 cancers-15-02065-f002:**
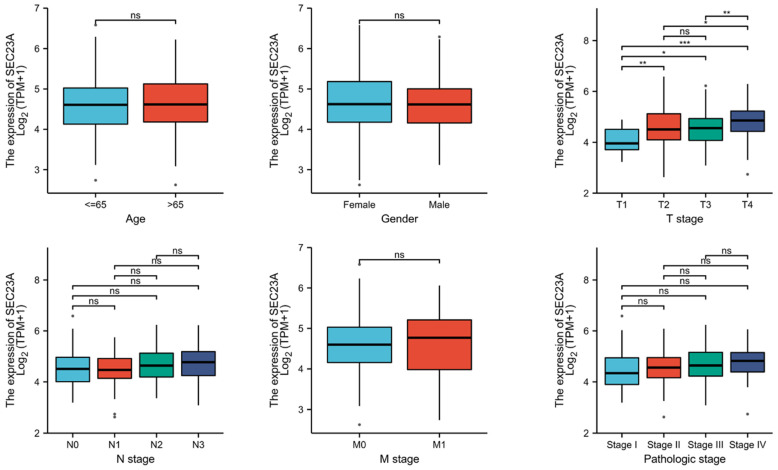
Relationship between clinicopathological variables and *SEC23A* expression (* *p* < 0.05, ** *p* < 0.01, *** *p* < 0.001, ns (not significant) *p* > 0.05).

**Figure 3 cancers-15-02065-f003:**
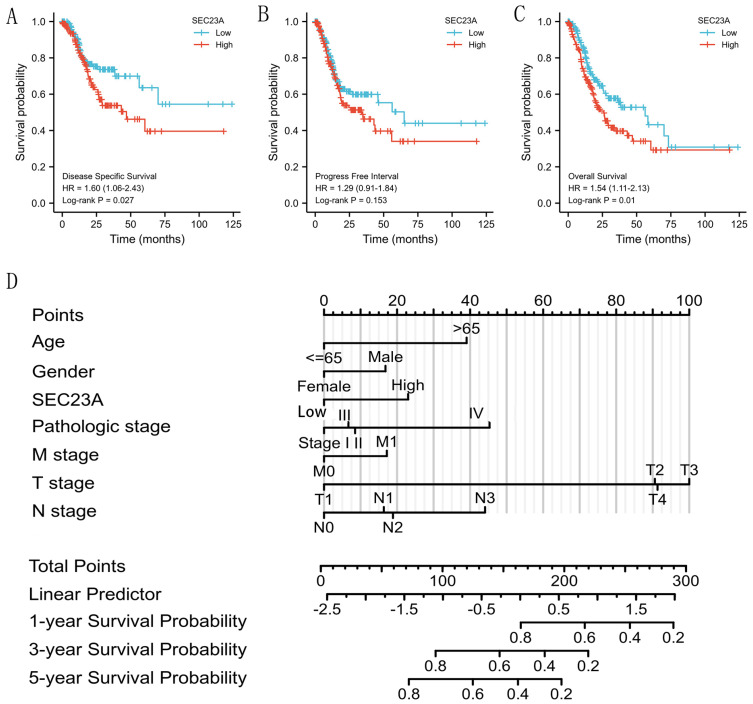
Association between *SEC23A* expression and prognosis. Increased *SEC23A* expression in STAD is associated with worse DSS (**A**), not with PFI (**B**) but again with worse OS (**C**). Multivariate Cox analysis of *SEC23A* expression and other clinicopathological factors (**D**). Points: for the single scores, they correspond to each predicted variable. Total Points: sum of the single score points. Linear Predictor: weighted sum of the variables in the Cox regression model with high values indicative of a worse prognosis.

**Figure 4 cancers-15-02065-f004:**
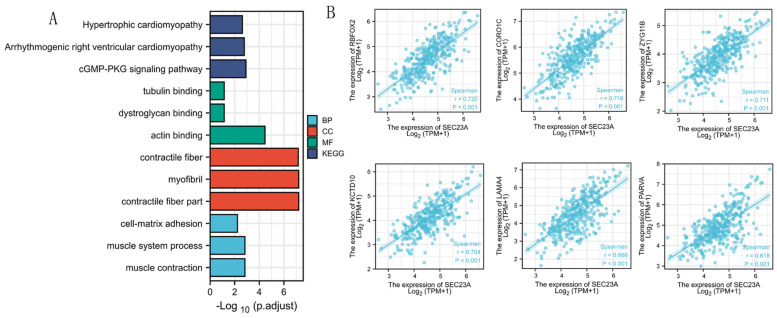
The top 100 genes co-expressed with *SEC23A* were selected to conduct an enrichment analysis. The results of GO biological process (BP) analysis, GO term cellular component (CC) analysis, molecular function (MF) and KEGG analysis are shown in (**A**). The association between *SEC23A* and the top 5 co-expressed genes (**B**).

**Figure 5 cancers-15-02065-f005:**
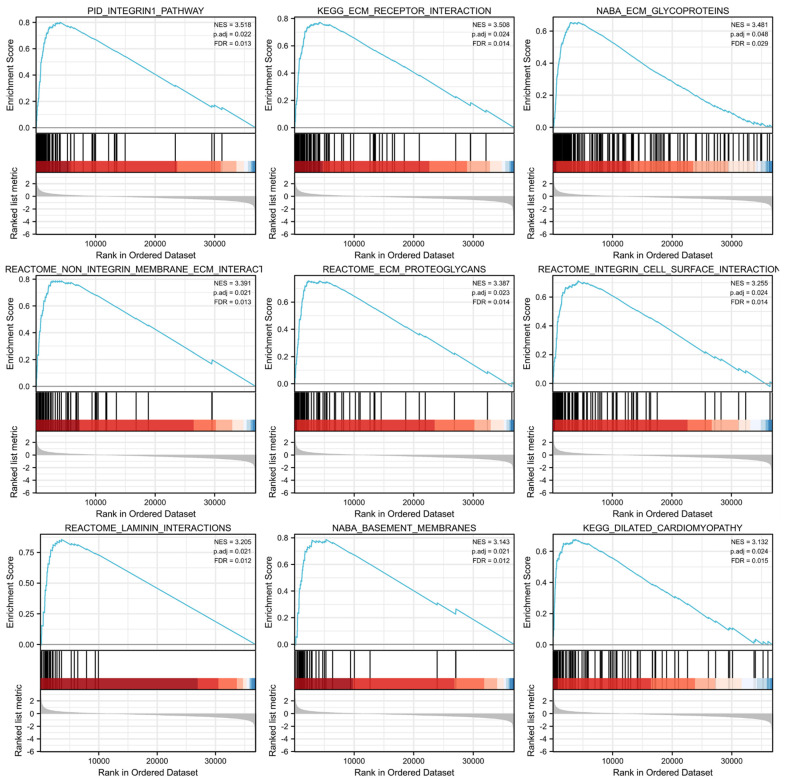
GSEA analysis of *SEC23A* in STAD.

**Figure 6 cancers-15-02065-f006:**
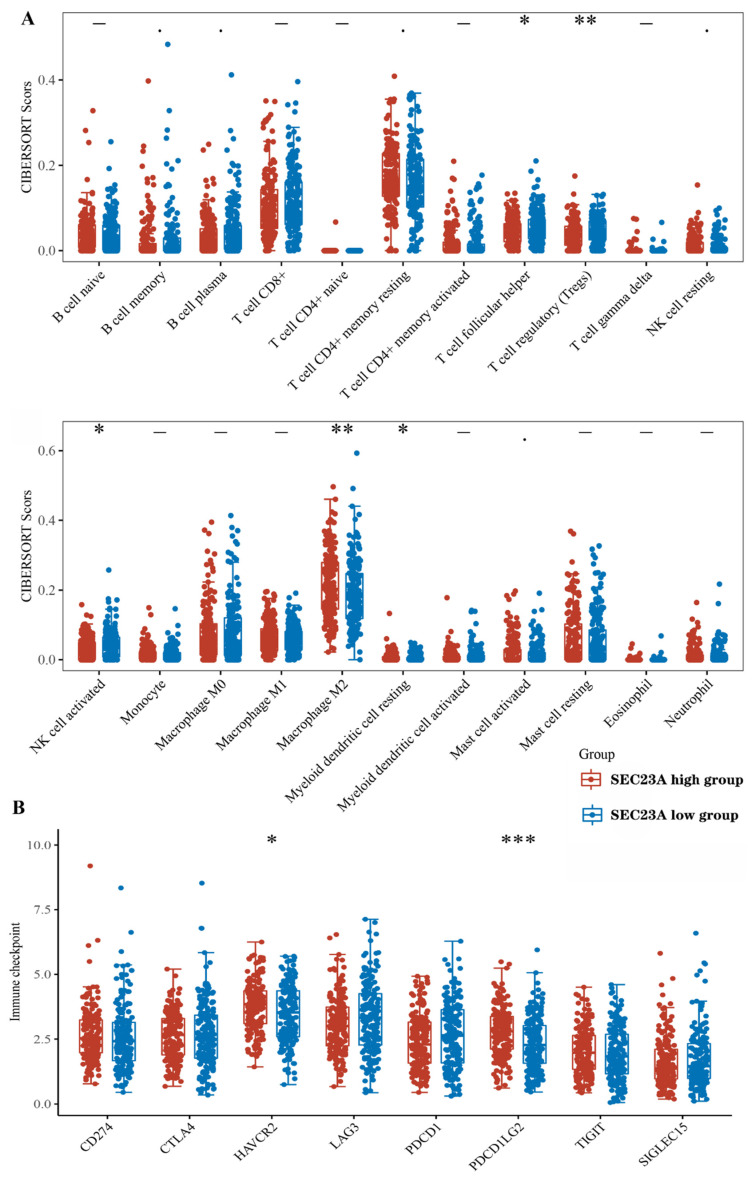
Tumor-infiltrating immune cell subsets and expression of checkpoint-related genes (* *p* < 0.05, ** *p* < 0.01, *** *p* < 0.001). Tumor-infiltrating immune cell subsets of the high and low *SEC23A* expression groups (**A**). Immune checkpoint molecule-related gene expression in the high and low *SEC23A* expression groups (**B**).

**Table 1 cancers-15-02065-t001:** Cox regression analysis for OS.

Characteristics	Total (N)	Univariate Analysis		Multivariate Analysis	
		Hazard Ratio (95% CI)	*p* Value	Hazard Ratio (95% CI)	*p* Value
T stage	347				
T1	18	reference			
T2	78	6.725 (0.913–49.524)	0.061	4.332 (0.548–34.233)	0.165
T3	157	9.548 (1.326–68.748)	0.025	4.984 (0.564–44.019)	0.148
T4	94	9.634 (1.323–70.151)	0.025	4.197 (0.460–38.322)	0.204
N stage	347				
N0	102	reference			
N1	97	1.629 (1.001–2.649)	0.049	1.299 (0.644–2.620)	0.465
N2	74	1.655 (0.979–2.797)	0.060	1.366 (0.581–3.212)	0.475
N3	74	2.709 (1.669–4.396)	<0.001	1.985 (0.845–4.664)	0.116
M stage	347				
M0	322	reference			
M1	25	2.254 (1.295–3.924)	0.004	1.216 (0.513–2.881)	0.658
Age	347				
≤65	153	reference			
>65	194	1.620 (1.154–2.276)	0.005	1.836 (1.264–2.667)	0.001
Gender	347				
Female	123	reference			
Male	224	1.267 (0.891–1.804)	0.188		
Pathologic stage	347				
UICC I	50	reference			
UICC II	110	1.551 (0.782–3.078)	0.209	1.113 (0.392–3.159)	0.840
UICC III	149	2.381 (1.256–4.515)	0.008	1.119 (0.284–4.415)	0.873
UICC IV	38	3.991 (1.944–8.192)	<0.001	2.204 (0.539–9.010)	0.271
*SEC23A*	347				
Low	184	Reference			
High	183	1.542 (1.106–2.151)	0.011	1.460 (1.016–2.098)	0.041

**Table 2 cancers-15-02065-t002:** Enrichment analysis of the top 100 genes co-expressed with SEC23A in STAD.

Ontology	ID	Description	GeneRatio	BgRatio	*p* Value	p.Adjust	q Value
BP	GO:0006936	muscle contraction	11/88	360/18,670	9.99 × 10^−7^	0.001	0.001
BP	GO:0003012	muscle system process	12/88	465/18,670	1.83 × 10^−6^	0.001	0.001
BP	GO:0007160	cell-matrix adhesion	8/88	225/18,670	1.10 × 10^−5^	0.006	0.005
CC	GO:0044449	contractile fiber part	12/91	221/19,717	4.22 × 10^−10^	6.16 × 10^−8^	4.75 × 10^−8^
CC	GO:0030016	myofibril	12/91	224/19,717	4.93 × 10^−10^	6.16 × 10^−8^	4.75 × 10^−8^
CC	GO:0043292	contractile fiber	12/91	234/19,717	8.13 × 10^−10^	6.77 × 10^−8^	5.22 × 10^−8^
MF	GO:0003779	actin binding	13/86	431/17,697	1.53 × 10^−7^	3.47 × 10^−5^	3.25 × 10^−5^
MF	GO:0002162	dystroglycan binding	2/86	10/17,697	0.001	0.071	0.067
MF	GO:0015631	tubulin binding	7/86	336/17,697	0.001	0.071	0.067
KEGG	hsa04022	cGMP-PKG signaling pathway	7/37	167/8076	8.70 × 10^−6^	0.001	0.001
KEGG	hsa05412	Arrhythmogenic right ventricular cardiomyopathy	5/37	77/8076	2.37 × 10^−5^	0.002	0.001
KEGG	hsa05410	Hypertrophic cardiomyopathy	5/37	90/8076	5.06 × 10^−5^	0.002	0.002

## Data Availability

Part of the data analyzed in this study were obtained from publicly available datasets. They can be found as described in materials and methods. Part of the data presented in this study are available in the article and the Appendix A.

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
