# Peer review of "Increased SEC23A Expression Correlates with Poor Prognosis and Immune Infiltration in Stomach Adenocarcinoma†"

_cancers, 2023, doi:10.3390/cancers15072065_

Round 1

Reviewer 1 Report

The paper titled "Increased SEC23A Expression Correlates with Poor Prognosis and Immune Infiltration in Stomach Adenocarcinoma" is interesting, but there are some major concerns that need to be addressed prior to publication.

Specific comments:

  1. The paper is missing important clinical characteristic tables for the datasets used from GEO, TCGA, and the in-house dataset. It is crucial to include detailed patient information, including age, gender, tumor stage, and other clinical parameters;
  2. The study findings have not been validated by experiments, which is a significant concern. While bioinformatic analysis can provide valuable insights, the results should be validated by experimental approaches, such as knockdown or overexpression of SEC23A in cancer cell lines to partially address the oncogenic function of this gene;
  3. Necessary references are missing, ie, Line 77-80;
  4. Please provide an explanation for reference 11 on Line 77;
  5. Figure 1G lacks statistical analysis. Please provide statistical analysis for this figure.
  6. Figure 2 shows no significant difference. Since two clinicopathological variables (T stage and pathologic stage) showed no consistency, please discuss this issue or consider moving this figure to supplementary materials.
  7. Figure 3D needs to be made more clear to read, as some sections are overlapping. Please revise the figure to improve its readability.
  8. On line 212, please provide a comment on the enriched pathways, not just describe the result.
  9. Please provide some explanations on the GO, KEGG, and GSEA enrichment analysis. Specifically, do these findings address the potential function of this gene, or are they only relevant to its association with other drivers or oncogenes? The findings from the GO, KEGG, and GSEA enrichment analysis can provide valuable insights into the potential function of this gene. However, further experimental validation is needed to confirm the function of this gene and its potential involvement in oncogenesis. It is important to note that the enriched pathways identified through the analysis provide a starting point for further investigation and hypothesis generation.
  10. Figure 5, please show the two groups used in the plot to provide better context for the data.

Author Response

The paper titled "Increased SEC23A Expression Correlates with Poor Prognosis and Immune Infiltration in Stomach Adenocarcinoma" is interesting, but there are some major concerns that need to be addressed prior to publication.

Re: We appreciate the time and effort the reviewer spent on the analysis of our paper. The comments are valuable and very helpful in revising and improving our paper. We have carefully studied the comments and have accordingly revised the manuscript.

Specific comments:

  1. The paper is missing important clinical characteristic tables for the datasets used from GEO, TCGA, and the in-house dataset. It is crucial to include detailed patient information, including age, gender, tumor stage, and other clinical parameters;

Re: We fully agree with the reviewer (as well as reviewer #3) that clinical characteristic data help readers in better understanding and interpreting the presented data. We added the background description of the clinical samples in the revised draft, accordingly and as complete as possible. However, not every sample of public database contain patient's complete clinical background information. So, we also submitted a flowchart in the supplementary material about the screening method for data of cases taken from public databases in the revised draft. We are confident, that this will make our work clearer to future readers. For the public database data, we detailed the background information including age, gender, tumor stage, and other clinical parameters of 347 patients incorporated into the COX regression analysis presented in Table 1.

We want to thank this reviewer for this specific request; it indeed strongly improved the overall quality of our analysis.

  1. The study findings have not been validated by experiments, which is a significant concern. While bioinformatic analysis can provide valuable insights, the results should be validated by experimental approaches, such as knockdown or overexpression of SEC23A in cancer cell lines to partially address the oncogenic function of this gene;

Re: Again, we absolutely agree with the reviewer´s comment that our findings demand experimental validation. Since this requires not only suitable experimental models (cell lines and/or animal models) but also molecular biological tools to manipulate these models, it is not feasible in a few weeks or even months. We started a follow-up project taking advantage of our large collection of patient-individual cancer cell lines. The results of these experiments will, however, be topic of a separate manuscript.

  1. Necessary references are missing, ie, Line 77-80;

Re: We added the necessary references and links in the revised version of our manuscript as the reviewer suggested.

  1. Please provide an explanation for reference 11 on Line 77;

Re:This explanation has been added.

  1. Figure 1G lacks statistical analysis. Please provide statistical analysis for this figure.

Re:This is indeed a witty comment. We performed a statistical analysis for the immunohistochemical staining results and added these data to the results part in the revised version of our manuscript. However, since we present in Figure 1G only representative stainings, adding it to the Figure legend, would not be beneficial.

  1. Figure 2 shows no significant difference. Since two clinicopathological variables (T stage and pathologic stage) showed no consistency, please discuss this issue or consider moving this figure to supplementary materials.

Re:Indeed, this inconsistency might be a remarkable finding. Significant difference was only found for the T stage variable - but we observed a trend for TNM. Since we wanted to show readers the relationship between clinicopathological variables and SEC23A expression, we followed the advice by keeping this figure and discussing this observation in the revised version of the manuscript.

  1. Figure 3D needs to be made more clear to read, as some sections are overlapping. Please revise the figure to improve its readability.

Re: As suggested, we redrew this figure to improve its readability. Additionally, we added information to the Figure legend to clarify the results.

  1. On line 212, please provide a comment on the enriched pathways, not just describe the result.

Re: This is, again, a very important comment. However, we are totally aware of the fact that it is typical for this type of analysis, that not all pathways found enriched easily fit into a clear picture. Thus, we took the freedom to discuss some possible connections between SEC23A and the enriched pathways found instead of just commenting in the results part.

  1. Please provide some explanations on the GO, KEGG, and GSEA enrichment analysis. Specifically, do these findings address the potential function of this gene, or are they only relevant to its association with other drivers or oncogenes? The findings from the GO, KEGG, and GSEA enrichment analysis can provide valuable insights into the potential function of this gene. However, further experimental validation is needed to confirm the function of this gene and its potential involvement in oncogenesis. It is important to note that the enriched pathways identified through the analysis provide a starting point for further investigation and hypothesis generation.

Re: This point is an extension of the previous one. As suggested and as mentioned before, we strengthened the discussion part by adding possible explanations on the results of the GO, KEGG, and GSEA enrichment analysis. Details can be found in the improved version of the manuscript – discussion part. We are confident, that this is convincing and also can be used a basis for further more functional investigations by our group as well as others.

  1. Figure 5, please show the two groups used in the plot to provide better context for the data.

Re: Fig 5 shows the results of GSEA in a standard graphical manner for this type of analysis. We agree with the reviewer that a scatterplot based on the two groups – high and low SEC23A expression - might be helpful to improve understanding of readers not familiar with this type of plot/data presentation. However, such a way of presenting the data has the disadvantage, that expression details, such as enrichment score, normalized enrichment score, false discovery rate and ranked list cannot be displayed as rich as the classic GSEA blots we used. Therefore, we kindly ask to accept, that this type of blot stays as it is.

Reviewer 2 Report

The manuscript entitled “ Increased SEC23A Expression Correlates with Poor Prognosis and Immune Infiltration in Stomach Adenocarcinoma “ by Zhaoran et al is a promising study in which SEC23A has been proposed as independent prognostic factor in STAD patients. They utilized computational methods as well as clinical samples to propose the same.

The study is well designed and discussed. One of the merits of the manuscript is that the limitations of the study is clearly explained.

An animal (mouse) data would be better to justify their speculations. However, They can plan it for their next communication(s).

Author Response

The manuscript entitled Increased SEC23A Expression Correlates with Poor Prognosis and Immune Infiltration in Stomach Adenocarcinoma by Zhaoran et al is a promising study in which SEC23A has been proposed as independent prognostic factor in STAD patients. They utilized computational methods as well as clinical samples to propose the same.

The study is well designed and discussed. One of the merits of the manuscript is that the limitations of the study is clearly explained.

An animal (mouse) data would be better to justify their speculations. However, They can plan it for their next communication(s).

Re: We appreciate the time and effort this reviewer spent on our paper and are grateful for his or her high opinion regarding our work. Moreover, we fully agree that the results of functional cell and/or animal experiments are required to verify some of our findings concerning the function of SEC23A.

Reviewer 3 Report

Topic of presented manuscript is suitable for the cancers. Design of studies, execution and evaluation of experiments is done well. Introduction is little to short, for example role of SEC23A the in other carcinogenesis should be mentioned. Possible limitation of this study could be also lover number of clinical participants, should be mentioned and discussed in discussion.

Author Response

Topic of presented manuscript is suitable for the cancers. Design of studies, execution and evaluation of experiments is done well.

Re: We want to thank this reviewer for this supportive comment.

Introduction is little to short, for example role of SEC23A the in other carcinogenesis should be mentioned.

Re: We modified and improved the manuscript according to this suggestion.

Possible limitation of this study could be also lover number of clinical participants, should be mentioned and discussed in discussion.

Re: We agree with this. However, since it is obvious to any future reader, we mentioned this fact together with the other limitations of this study. In future work, which is currently ongoing for colorectal cancer, a larger number of clinical cases is investigated. In addition, we will there also add functional analyses.

Reviewer 4 Report

In this paper the authors propose SEC23A as independent prognostic factor with a putative role with immune response regulation in STAD but there is no report in PubMed showing SEC23A alterations in human cancer; the role of SEC23A in human cancer has been reported as a target of other genes, like miRNA (Ventura A, Jacks T. MicroRNAs and cancer: short RNAs go a long way. Cell. 2009; 136:586–91); thus, it remains to be determined how SEC23A regulates the development of human cancer (Korpal M, Ell BJ, Buffa FM. et al. Direct targeting of Sec23a by miR-200s influences cancer cell secretome and promotes metastatic colonization. Nat Med. 201;17:1101–8.). Any authors reported that the Knockdown of SEC23A expression enhanced tumor cell proliferation and metastatic colonization in prostate and colorectal cancer. It was possibly because SEC23A could mediate secretion of Insulin-Like Growth Factor Binding Protein 4 (IGFBP4), a metastasis-suppressive protein. Interestingly, SEC23A has a clear inhibitory role in breast cancer metastasis, especially the step of colonization during tumor cell metastasis but not at the step of tumor cell migration. In this scenario, we advise the authors to correlate what they said to the evidence reported in the literature.

The introduction is brief and incomplete, it does not report all the information present on SECA and the two isoforms of its proteins. it is recommended to expand.

We suggest clarifying the choice of sample for the analysis of poor prognosis, inserting a table that clarifies TNM, population etc... enter analysis of OV.

Based on the evidence of paper 10.3389/fgene.2021.672832, we ask the authors to clarify the differences between the two papers and to clarify to the reader where possible the different applications. SEC23A appears to be too nonspecific to be a prognostic factor.

Author Response

In this paper the authors propose SEC23A as independent prognostic factor with a putative role with immune response regulation in STAD but there is no report in PubMed showing SEC23A alterations in human cancer; the role of SEC23A in human cancer has been reported as a target of other genes, like miRNA (Ventura A, Jacks T. MicroRNAs and cancer: short RNAs go a long way. Cell. 2009; 136:58691); thus, it remains to be determined how SEC23A regulates the development of human cancer (Korpal M, Ell BJ, Buffa FM. et al. Direct targeting of Sec23a by miR-200s influences cancer cell secretome and promotes metastatic colonization. Nat Med. 201;17:11018.). Any authors reported that the Knockdown of SEC23A expression enhanced tumor cell proliferation and metastatic colonization in prostate and colorectal cancer. It was possibly because SEC23A could mediate secretion of Insulin-Like Growth Factor Binding Protein 4 (IGFBP4), a metastasis-suppressive protein. Interestingly, SEC23A has a clear inhibitory role in breast cancer metastasis, especially the step of colonization during tumor cell metastasis but not at the step of tumor cell migration. In this scenario, we advise the authors to correlate what they said to the evidence reported in the literature.

Re: We thank this reviewer for the constructive comment, allowing also the interpretation, that he or she has a very rich research background on SEC23A. Moreover, we agree that previous studies have associated SEC23A with miRNA (miRNA 375, 200c/s, 21, 29b-3p, etc.) and IGFBP4. Judging from our own preliminary results as well as many published studies, the impact of SEC23A on tumor progression and metastization is likely manifold. For example, it seems to vary for different tumor entities. Currently, we are working on its role in the context of tumor-immune interaction, apoptosis and drug resistance.

To come to the point: based on the reviewer's suggestion, we modified the discussion part on a large scale. Details can be depicted from the improved version of the manuscript.

The introduction is brief and incomplete, it does not report all the information present on SECA and the two isoforms of its proteins. it is recommended to expand.

Re: We thank this reviewer for the hint. Accordingly, we expanded the introduction and added information on SEC23 including the two isoforms of its proteins in the revised draft.

We suggest clarifying the choice of sample for the analysis of poor prognosis, inserting a table that clarifies TNM, population etc... enter analysis of OV.

Re: We fully agree with this reviewer (as well as reviewer #1) that clinical characteristic data help readers in better understanding and interpreting the presented data. We added the background description of the clinical samples in the revised draft, accordingly and as complete as possible. Not every sample of public database contain patient's complete clinical background information, and we did not choose samples, but included all with sufficient information for the single analyses. To make that process clear, we also submitted a flowchart in the supplementary material about the screening method for data of cases taken from public databases in the revised draft. We are confident, that this will make our work clearer to future readers. For the public database data, we detailed the background information including age, gender, tumor stage, and other clinical parameters of 347 patients incorporated into the COX regression analysis presented in Table 1.

We want to thank this reviewer for this specific request; it indeed strongly improved the overall quality of our analysis.

Based on the evidence of paper 10.3389/fgene.2021.672832, we ask the authors to clarify the differences between the two papers and to clarify to the reader where possible the different applications. SEC23A appears to be too nonspecific to be a prognostic factor.

Re: We thank this reviewer for the constructive comment. The discussion part has been adapted accordingly in the revised version of the draft.

Round 2

Reviewer 1 Report

The authors were able to address part of my concerns successfully, and I have no additional comments to provide.